# Geostatistical Analysis of Soil C/N Deficiency and Its Effect on Agricultural Land Management of Major Crops in Eastern Croatia

**Mladen Jurišić [1], Dorijan Radočaj [1,*] , Stjepan Krčmar [2], Ivan Plaščak [1] and Mateo Gašparović [3]**

[1]  Faculty of Agrobiotechnical Sciences Osijek, Josip Juraj Strossmayer University of Osijek, Vladimira Preloga 1, 31000 Osijek, Croatia; mjurisic@fazos.hr (M.J.); iplascak@fazos.hr (I.P.)

[2]  Department of Biology, Josip Juraj Strossmayer University of Osijek, Cara Hadrijana 8/A, 31000 Osijek, Croatia; stjepan@biologija.unios.hr

[3]  Faculty of Geodesy, University of Zagreb, Kačićeva 26, 10000 Zagreb, Croatia; mgasparovic@geof.unizg.hr

*  Correspondence: dradocaj@fazos.hr; Tel.: +385-31-554-879

**Abstract:** Interpolation accuracy of the soil carbon-to-nitrogen ratio (C/N) has a direct impact on soil maps for agricultural land management planning, affecting the financial gains of farmers and environmental contamination. This study aimed to evaluate the effect of different ordinary kriging parameters on the interpolation accuracy and spatial variability of soil C/N. These values were necessary to determine the effect of potential soil C/N deficiency on five major crops in the study area during 2017–2019, containing maize, wheat, sunflower, rapeseed and soybean parcels. The Gaussian model with logarithmic transformation produced the higher outer accuracy, representing the prediction accuracy. Average values of final soil C/N rasters resulted as 11.35 at 0–10 cm soil depth and 12.60 at 20–30 cm soil depth, representing values 11.0% higher at a deeper soil layer. The moderate soil N deficiency was observed for all major crops, having soil C/N values mainly higher than 12:1, compared to the optimal value range of 9:1 to 12:1. These observations suggest a need for the adjustment of agricultural land management plans through the regionalization of agricultural production to prevent further soil degradation and ensure optimal crop development conditions.

**Keywords:** soil degradation; GIS; ordinary kriging; organic farming; sustainable agriculture; regionalization; organic matter

## 1. Introduction

The cultivation of agricultural soils inevitably accelerates the processes of humus decomposition, which results in the reduction of organic matter content [1]. The rate of organic matter content reduction is dependent on the land management system, and every agrotechnical operation should consider the amount of soil organic matter [2]. Soil organic matter represents a major source of most nutrients and consists of significant quantities of carbon (C) in combination with smaller quantities of nitrogen (N) [3]. The soil carbon-to-nitrogen ratio (C/N) has a direct impact on the agricultural residues' degradation, as well as on the circulation of N in soils [4]. Climate changes with the increase of annual air temperatures and lower precipitation indirectly stimulate soil organic carbon mineralization, obstructing the input of soil organic carbon and gradually decreasing its quantity [5]. Soil C/N is one of the primary indicators of these effects, reflecting the severity of soil degradation as the lower capacity of soil for transfer of important nutrients to vegetation and microorganism development [6]. The intensive agricultural production caused by the growing necessity for higher crop quantity demands invariably

affects dynamics of carbon and nitrogen, endangering the sustainability of present cropping systems [7]. Recent research by Bogunović et al. [8] supported the claim that the intensive agricultural production negatively influences these dynamics, specifically reducing soil organic matter and soil organic carbon stocks. To ensure the sustainability of agricultural production in such areas, fertilization, weed control and soil tillage methods should be included in agricultural land management plans, as they directly or indirectly affect long-term soil degradation [9,10]. Furthermore, the regulation of soil C/N is necessary for optimal agricultural land management and the regionalization of crop production, having a major impact on crop development and growth [11]. This is especially relevant for organic farming, as it mainly depends on the circulation of soil organic matter through crop residues within an agricultural parcel [12]. Crop yield produced by organic farming is generally lower comparing to the conventional approach, which reinforces the need for soil C/N management for sustainable organic farming [13]. As per Eurostat data [14], the average percentage of total agricultural area utilized for organic farming was 7.50% in 2019, which steadily increased compared to 5.64% in 2012. There is still a large heterogeneity of organic farming policies within the European Union (EU), as these percentages during 2018 vary from 2.43% for Romania and 24.08% for Austria. Croatia in among the EU countries with the highest trend of organic farming expansion, growing from 2.40% in 2012 to 6.94% in 2018 of total agricultural area utilized for organic farming and is expected to surpass the EU average in forthcoming years.

Reliable soil maps are fundamental decision-making bases for soil C/N management [15]. A Geographic Information System (GIS) combines the tools for collecting and processing the field samples, their processing by geostatistical methods, geospatial analysis and the creation of thematic maps [16]. These data are commonly implemented in various suitability analyses as a part of GIS-based multicriteria analysis studies for agricultural and environmental purposes [17]. In soil analysis, it is possible to collect only a limited number of samples, which in such form do not give a full representation of soil properties. The spatial variability of these soil values should also be considered, which indicates a necessity for spatial interpolation methods [18]. Geostatistical methods enable the interpolation of values between sample points used for soil map creation [19]. Soil maps are vital for precision farming, and their quality depends on the quality and density of sample points and interpolation parameters [20]. Kriging was proven to be superior in most cases to deterministic interpolation techniques for interpolation of soil chemical properties [21]. Ordinary kriging (OK) is the most commonly used kriging method and was successfully applied for the interpolation of various soil properties [22,23]. Normality and stationarity of input data are two requirements for the interpolation using OK [24]. These properties, as well as other properties like the number of samples and value ranges, determine the optimal interpolation parameters and interpolation accuracy [25]. Therefore, an evaluation of interpolation parameters is necessary for accurate creation of soil maps [26].

The hypothesis of the study is that the properties of input samples and interpolation parameters have a major impact on interpolation accuracy and the predicted spatial variability of soil C/N using OK. The specific objectives of the study were to: (1) evaluate normality, stationarity and other relevant properties of input samples, (2) perform spatial interpolation using OK with different interpolation parameters and assess their accuracy, and (3) evaluate the areas of soil C/N deficiency and determine its effect on major crops in the study area.

## 2. Materials and Methods

### 2.1. Study Area and Data Acquisition

The study area was the County of Osijek-Baranja, situated in eastern Croatia (Figure 1). Agricultural land covers 2082 km² (50.1%) of the total 4155 km² county area, according to the Paying Agency for Agriculture, Fisheries and Rural Development data for the year 2019. As reported by the Croatian Bureau of Statistics in 2016, 14% of Croatia's agricultural area is being utilized in the County of Osijek-Baranja, this being the traditionally first-ranked county in agricultural production. According

to the same source, more than 20% of cereals, 46% of sugar beet and 17% of orchards from the total country's agricultural area were cultivated in County. The recent crop suitability studies in the study area [25,27] showed that the current agricultural land management plans could be upgraded to allow considerably better regionalization of agricultural production for individual crops. The update of obsolete agricultural land management according to climate changes is crucial for a stable crop yield and the potential export of high-quality crops [28]. The traditionally intensive conventional tillage in eastern Croatia, with ploughing as the main tillage procedure, mainly affects soil organic matter [29]. As per the European soil database v2.0 (https://esdac.jrc.ec.europa.eu/content/european-soil-database-v20-vector-and-attribute-data), orthic luvisol and calcaric fluvisol are the most common soil types, covering 59.3% of the study area combined. A moderately warm and rainy climate type class 'Cfwbx' is present in Osijek-Baranja County, classified according to a Köppen scale [30]. Historical climate data (1899–2018) for the weather station Osijek near the centroid of the county showed an average annual air temperature of 11.1 °C and the average total annual precipitation of 692.9 mm [31]. The average summer air temperature calculated from June to September during the same period was 19.8 °C.

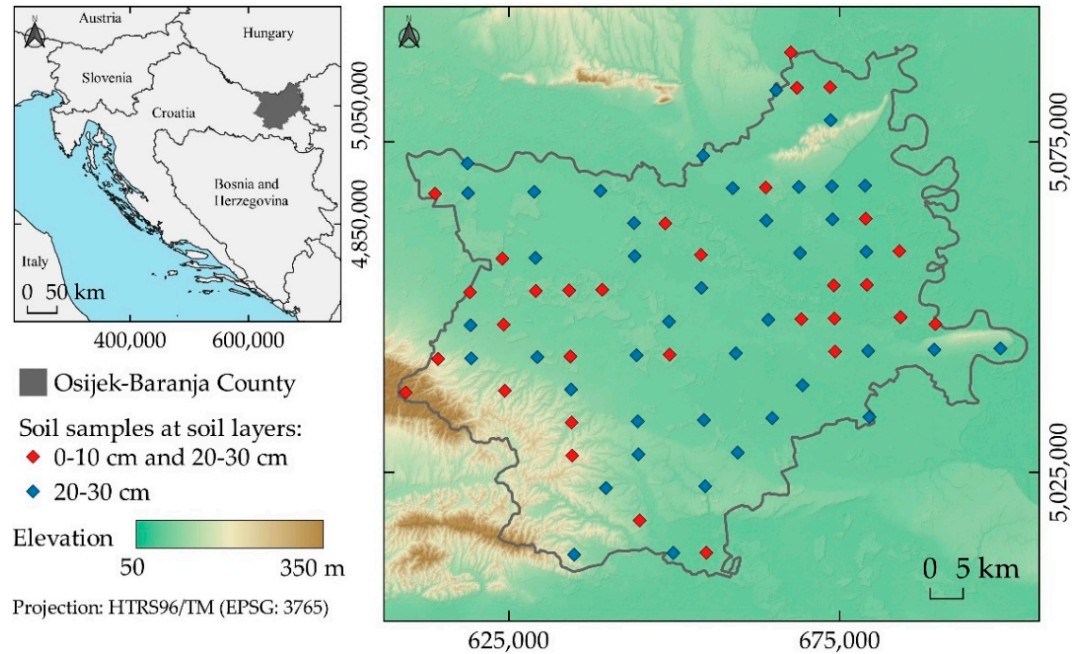

**Figure 1.** The location of Osijek-Baranja County and soil samples at 0–10 and 20–30 cm soil layers.

Four sample data sets were obtained from the Croatian Agency for the Environment and Nature soil data WFS service, collected in the field during the year 2016. Pre-treatment of soil samples was conducted according to ISO 11464:2006, while the elemental analysis using dry combustion for soil C and N was performed according to ISO 10694:1995 and ISO 13878:1998. A total of 31 samples were collected in the agricultural land on 0–10 cm soil depth for C (C 0–10 cm) and N (N 0–10 cm) and 64 samples on 20–30 cm soil depth for C (C 20–30 cm) and N (N 20–30 cm). These samples were collected as the supplementary data to the national soil sampling in 2013, causing an uneven sampling distribution at the 0–10 cm and 20–30 cm soil layers. Soil sampling at the 10–20 cm soil layer was omitted from the process, which is the reason why it is not contained in the input dataset. Soil sample values labeled as 'C' designate soil organic carbon and 'N' designate total soil nitrogen. These values were successfully implemented for the evaluation of soil C/N properties in [32,33]. The measurement unit for both soil C and N was g 100 g$^{-1}$. These sampling depths were selected as soil C/N generally remains consistent with the increase of soil depth and is mainly influenced by the crops, while about of 80% of the soil organic carbon is located at the 0–40 cm soil depth [34].

Agricultural parcel data was received by the Croatian Paying Agency for Agriculture, Fisheries and Rural Development from 2017, 2018 and 2019 for five major crops in Osijek-Baranja County. Maize, wheat, sunflower, rapeseed and soybean consistently amounted to about two thirds of total county cultivated crop area, having a major importance on the county's economy. Previous studies proved that soil C/N values indicate the yield of maize and wheat [32], sunflower [35], rapeseed [36] and soybean [37]. Annual cultivated area and the part of total county agricultural area per crop are displayed in Table 1. The agricultural parcels used were visualized using three example subsets, as shown in Figure 2. These example subsets cover three representative areas regarding the land-cover and land-use systems: a western part consisting of fragmented agricultural parcels surrounded by forests, a central part characterized by the intensive agricultural production, and an eastern part in Baranja region with large parcels of a major agricultural company in the proximity of wetland. The total cultivated agricultural area in the study area was 209,874 ha in 2017, 209,702 ha in 2018 and 208,163 ha in 2019.

**Table 1.** Annual crop coverage statistics of five major crops in the Osijek-Baranja County.

| Crop | 2017 | | 2018 | | 2019 | |
|---|---|---|---|---|---|---|
| | Area Cultivated (ha) | Part of Total Agricultural Area (%) | Area Cultivated (ha) | Part of Total Agricultural Area (%) | Area Cultivated (ha) | Part of Total Agricultural Area (%) |
| Maize | 42,368 | 20.19 | 40,232 | 19.19 | 56,549 | 27.17 |
| Wheat | 32,941 | 15.70 | 40,486 | 19.31 | 42,048 | 20.20 |
| Sunflower | 22,347 | 10.65 | 23,818 | 11.36 | 20,179 | 9.69 |
| Rapeseed | 20,981 | 10.00 | 21,039 | 10.03 | 14,137 | 6.79 |
| Soybean | 16,642 | 7.93 | 15,117 | 7.21 | 13,199 | 6.34 |
| Total | 135,279 | 64.46 | 140,692 | 67.09 | 146,112 | 70.19 |

### 2.2. Geostatistical Analysis

The geostatistical analysis was conducted in three steps: evaluation of normality and stationarity of input samples, evaluation of properties and accuracy of the interpolation results, and interpolation of soil C/N with the assessment of spatial variability.

Evaluation of input samples consisted of tested for data normality and stationarity, as required assumptions for kriging [24]. The normality test was used for the assessment of sample values distribution, whereas stationarity was used for the assessment of spatial variability distribution. Descriptive statistics consisting of the mean, minimum, maximum, coefficient of variation (CV), skewness and kurtosis were used for the evaluation of data normality and stationarity. The distribution of data was additionally examined using the Shapiro–Wilk normality test [38]. A borderline *p*-value of 0.05 was used for the evaluation of the null hypothesis of normal data distribution with 95% confidence. The Shapiro–Wilk test was performed using Python version 3.7.3 (Scotts Valley, CA, USA) with library SciPy version 1.3.0. Stationarity (Austin, TX, USA) was evaluated by Thiessen polygons, indicating differences in local variations of the sample data. Thiessen polygons were also used to indicate locations of sparse sample data, having notably larger area polygons than others.

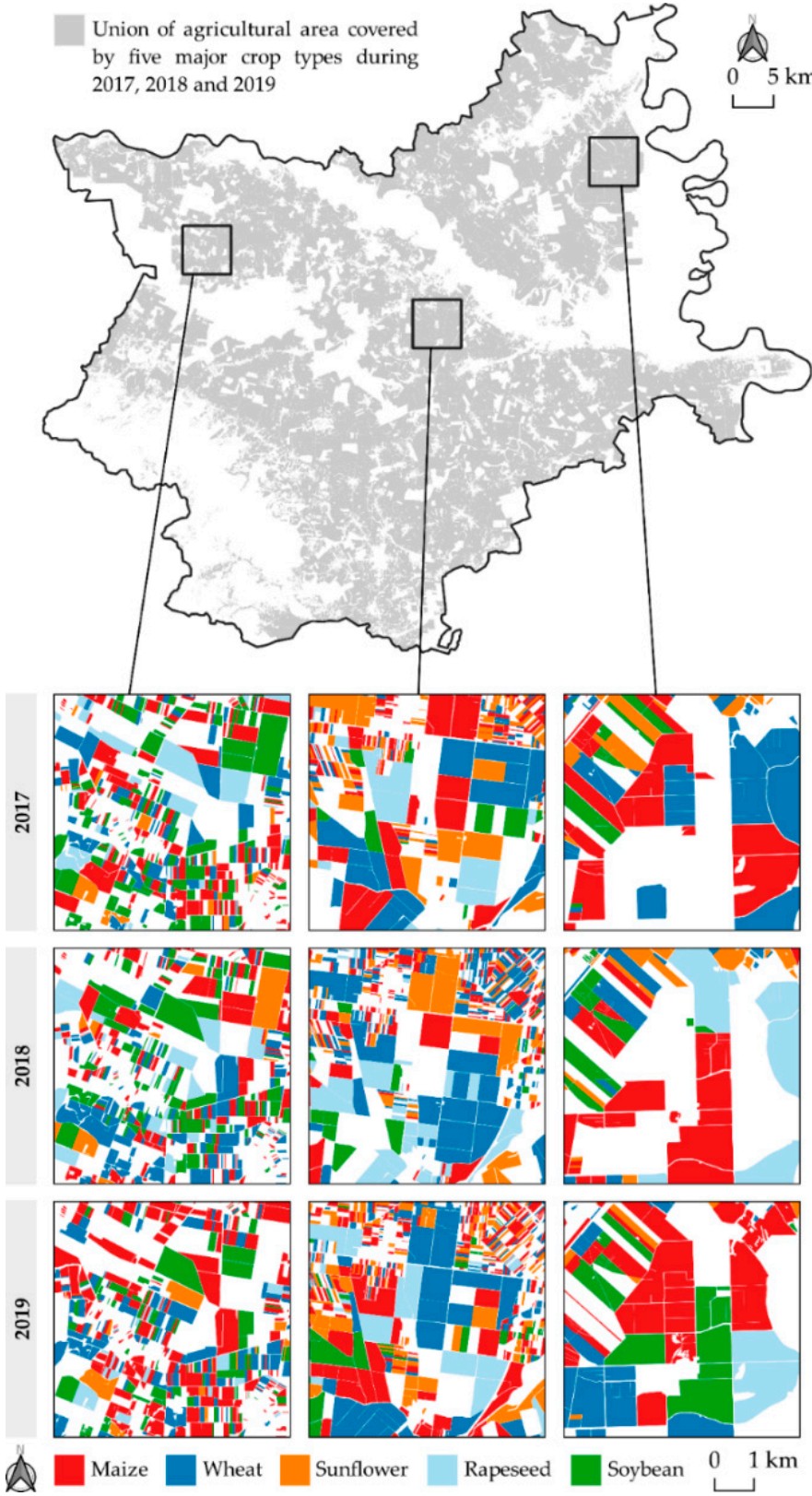

**Figure 2.** Annual area covered by five major crops in the Osijek-Baranja County during 2017–2019 and the three enlarged example subsets.

Interpolation was performed using the OK method, which is based on the calculation of spatial autocorrelation using the variogram calculated according to the following formula (Equation (1)) [19]:

$$\gamma(h) = \frac{1}{2m(h)} \sum_{i=1}^{n(h)} [z(x_i) - z(x_i+h)]^2,$$ (1)

where $\gamma(h)$ is the semivariance, $z(x_i)$ and $z(x_i + h)$ are the input values at locations $x_i$ and $x_i + h$, $n(h)$ is the number of paired comparisons at distance $h$. Evaluated mathematical models for the fitting to empirical variogram were linear, square root, Gaussian and spherical [39]. The formula for the prediction of values based on OK method was (Equation (2)) [19]:

$$z(x) = \sum_{i=1}^{n} \lambda_i z(x_i),$$ (2)

where $z(x)$ is predicted value at location $x$ and $\lambda_i$ is the weight calculated using the variogram. The interpolation using logarithmic transformation was evaluated in the research, being recommended in the absence of normal distribution of input data [40]. Each of the four mathematical models was evaluated with and without logarithmic transformation for the quantification of the impact of a normal distribution on spatial variability. A total of 9 lags were used in the interpolation for 0–10 samples, while 12 lags were applied for the interpolation using 20–30 cm samples. Geostatistical calculations were conducted in SAGA GIS version 7.3.0 (Hamburg, Germany). The spatial resolution of all interpolation results was set to 250 m.

Nugget (*n*), sill (*s*) and range (*r*) values were calculated for each iteration using different OK interpolation parameters for the determination of interpolation characteristics. The degree of spatial dependence was calculated for an indication of the spatial autocorrelation strength. Soil C/N rasters were calculated by dividing interpolated C and N rasters of the corresponding interpolation parameters. By combining the four evaluated input samples, mathematical models and logarithmic transformation, a total of 16 interpolation variants using OK were calculated. Descriptive statistics consisting of mean, minimum, maximum and CV were calculated for the final soil C/N values to quantify value ranges and spatial variability produced by various parameters.

### 2.3. Accuracy Assessment of Interpolated Results

Coefficients of determination were used for the accuracy assessment, representing inner ($R_I^2$) and outer accuracy ($R_O^2$). This procedure was successfully applied in a study by Jurišić et al. [26]. Inner accuracy showed the correlation between input sample values and interpolation results on an identical location. This value quantified the preservation of the ground-truth values in the interpolation results. Outer accuracy represented the prediction accuracy of the particular interpolation parameters at an unknown location. Outer accuracy was evaluated using a cross-validation method with a 2-fold procedure. The calculation was performed by the interpolation with missing random two samples per iteration and repeating the same procedure until all of the input samples were evaluated.

### 2.4. Soil Carbon-to-Nitrogen Ratio (C/N) Deficiency Evaluation for Five Major Crops during 2017–2019

Soil C/N interpolation results which produced the highest outer accuracy were determined as representative and selected for further analysis. The evaluation of soil C/N dynamics for five major crops was performed for 2017 as soil sampling was conducted the previous year. This was achieved by the calculation of mean and standard deviation values for each of five major crops, covering all agricultural parcels within the study area of a particular crop. The analysis was performed individually for 0–10 cm and 20–30 cm soil depths, as the 0–10 cm soil layer is particularly susceptible to soil degradation and reflects the areas of the most intensive agricultural production [41]. Based on the same soil C/N interpolation results, the projections for 2018 and 2019 were created using the same approach

to determine the suitability of current crop management plans according to the present state for soil degradation. The soil C/N ratios in ranges of 10:1 to 12:1 are regarded as the standard values for the agriculturally important cultivated soils, allowing an optimal balance of carbon oxidation process and nitrogen immobilization [42]. The soil C/N of intensely cultivated soils is generally expected to produce lower soil C/N compared to the uncultivated soils, based on the same source. The threshold values of soil C/N for the balance of these processes differed from 9.7 and 10.8, depending on the soil texture, C and N soil contents [43]. Marchetti et al. [44] accordingly proposed the soil C/N classification in three classes: optimal conditions ($9 \leq C/N \leq 12$), dominant oxidation (C/N < 9) and nitrogen deficiency (C/N > 12). The severity of soil degradation according to five major crops was evaluated according to the soil C/N classification in these classes.

## 3. Results

Higher input mean values for both C and N were observed in the top soil layer (Table 2). All samples possessed a moderate variability, with an average value of 0.53. High skewness values indicated a non-normal value distribution for all samples. Kurtosis values showed a platykurtic distribution, except for N 20–30 cm sample. Shapiro–Wilk test results suggested the absence of normal distribution of values for all input samples, as all of the *p*-values were below the border value. C 0–10 cm sample had the closest values distribution to normal, having the lowest skewness value and the highest *p*-value of observed samples. Both N samples produced very low W- and *p*-values.

**Table 2.** Descriptive statistics and Shapiro–Wilk normality test for input samples.

| Values | | C 0–10 cm | N 0–10 cm | C 20–30 cm | N 20–30 cm |
|---|---|---|---|---|---|
| Mean (g $100\ g^{-1}$) | | 2.58 | 0.24 | 1.98 | 0.16 |
| Minimum (g $100\ g^{-1}$) | | 0.97 | 0.09 | 0.74 | 0.04 |
| Maximum (g $100\ g^{-1}$) | | 5.88 | 0.66 | 5.42 | 0.48 |
| CV | | 0.48 | 0.52 | 0.59 | 0.53 |
| Skewness | | 0.95 | 1.50 | 1.34 | 1.87 |
| Kurtosis | | 0.53 | 1.82 | 1.26 | 3.61 |
| Shapiro–Wilk | W | 0.81 | 0.47 | 0.85 | 0.42 |
| | *p* | 0.029 | >0.001 | >0.001 | >0.001 |

Thiessen polygons suggested that all input samples possessed a moderate stationarity. The mild transition of neighboring colors indicated the lower local variation between the two polygons. Minor anomalies in the properties of Thiessen polygons were observed in the central area of the County, having a high local variation (Figure 3). The highest amounts of C and N are generally grouped on the eugley soils in the northeastern part of the study area. Areas with mainly cultivated agricultural land in the central part had the lowest values. Polygons for samples in 20–30 cm soil depth have an approximately even area through the County, indicating even uniform spatial distribution of samples. For samples in 0–10 cm soil depth, polygons have varying areas, the largest being in the central part.

Evaluation of properties and accuracy of the interpolation results produced by various interpolation parameters was performed according to statistical values for 0–10 cm (Table 3) and 20–30 cm samples (Table 4). Interpolation with no transformation produced the higher spatial dependence coefficient compared to interpolation with logarithmic transformation in 12 out of 16 interpolation variants. All four cases with a higher spatial dependence using logarithmic transformation occurred in highly skewed N samples. An overall average spatial dependence using logarithmic transformation was considerably higher for N samples (0.888), compared to C samples (0.572). A similar tendency occurred for both inner and outer accuracies since N samples produced a higher average $R_I^2$ of 11.5% and average $R_O^2$ of 9.2% compared to C samples with the application of logarithmic transformation. The difference of average spatial dependence using logarithmic transformation between N and C samples of 0.316 is considerably higher than in the case of using no transformation (0.178). Logarithmic transformation produced lower $R_I^2$ values compared to no transformation, with average respective values of 78.4%

and 87.4% for C samples, alongside 89.9% and 93.1% for N samples. However, the application of logarithmic transformation allowed a more accurate prediction of variables, having higher $R_O^2$ than no transformation of 8.5% for C and 22.9% for N samples. A spherical model produced a higher average $R_I^2$ compared to other mathematical models, both observing the chemical properties (86.9% for C and 96.5% for N) and soil sampling depths (97.4% for 0–10 cm and 86.0% for 20–30 cm). Similarly to the spherical model for inner accuracy, the Gaussian model resulted in the highest average $R_O^2$ for the chemical properties (56.7% for C and 61.9% for N) and soil sampling depths (58.5% for 0–10 cm and 60.1% for 20–30 cm). The effect of sparser sampling density was present in the outer interpolation accuracy values, with interpolation results of 20–30 cm samples having 3.2% higher average $R_O^2$ compared to results of the 0–10 cm samples. The variograms of mathematical models that produced the highest inner and outer accuracy per input sample are shown in Figure 4.

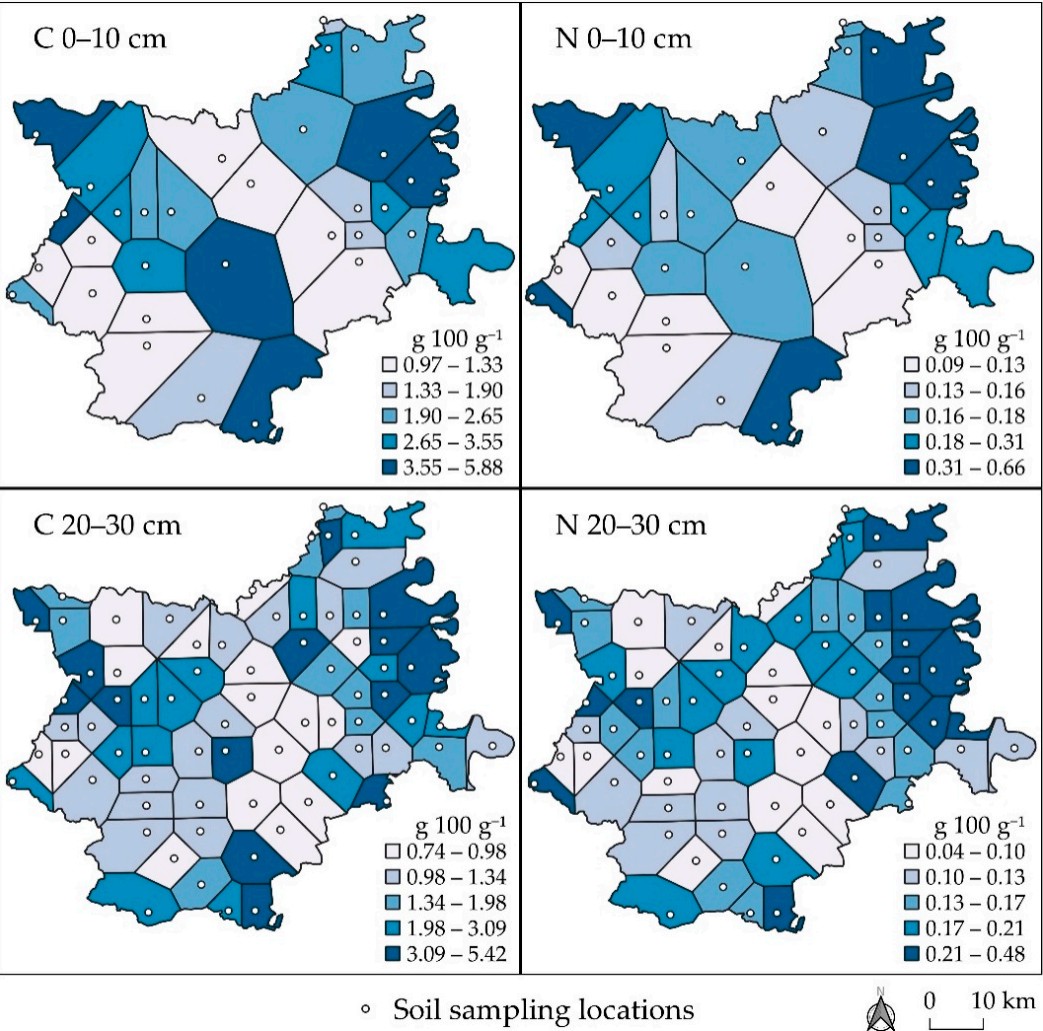

**Figure 3.** Thiessen polygons for the stationarity evaluation of input samples.

**Table 3.** Statistical properties of interpolation results for 0–10 cm soil depth.

| Soil Element | Model | Transformation Method | $n$ | $s$ | Spatial Dependence | $r$ (m) | $R_I^2$ (%) | $R_O^2$ (%) |
|---|---|---|---|---|---|---|---|---|
| C | linear | none | 0.359 | 2.367 | 0.848 | 25348 | 95.7 | 53.5 |
|  |  | logarithmic | 0.064 | 0.345 | 0.814 |  | 93.6 | 58.8 |
|  | square root | none | 0.298 | 2.159 | 0.862 | 37471 | 96.0 | 42.8 |
|  |  | logarithmic | 0.025 | 0.342 | 0.927 |  | 96.8 | 55.1 |
|  | Gaussian | none | 0.136 | 2.270 | 0.940 | 25348 | 97.1 | 58.7 |
|  |  | logarithmic | 0.041 | 0.330 | 0.876 |  | 95.0 | 59.5 |
|  | spherical | none | 0.098 | 1.798 | 0.945 | 24768 | 99.2 | 54.3 |
|  |  | logarithmic | 0.022 | 0.280 | 0.921 |  | 98.5 | 59.1 |
| N | linear | none | 0.037 | 0.188 | 0.803 | 42981 | 95.8 | 42.9 |
|  |  | logarithmic | 0.147 | 0.672 | 0.781 |  | 80.5 | 65.5 |
|  | square root | none | 0.035 | 0.135 | 0.741 | 55089 | 97.4 | 38.0 |
|  |  | logarithmic | 0.158 | 0.514 | 0.693 |  | 81.1 | 63.0 |
|  | Gaussian | none | 0.009 | 0.190 | 0.953 | 42981 | 93.8 | 49.5 |
|  |  | logarithmic | 0.071 | 0.649 | 0.891 |  | 88.1 | 66.3 |
|  | spherical | none | 0.003 | 0.100 | 0.970 | 36049 | 99.7 | 50.9 |
|  |  | logarithmic | 0.087 | 0.417 | 0.791 |  | 92.2 | 63.3 |

**Table 4.** Statistical properties of interpolation results for 20–30 cm soil depth.

| Soil Element | Model | Transformation Method | $n$ | $s$ | Spatial Dependence | $r$ (m) | $R_I^2$ (%) | $R_O^2$ (%) |
|---|---|---|---|---|---|---|---|---|
| C | linear | none | 0.705 | 1.536 | 0.541 | 21,194 | 72.0 | 57.5 |
|  |  | logarithmic | 0.220 | 0.297 | 0.259 |  | 54.5 | 56.2 |
|  | square root | none | 0.319 | 1.458 | 0.781 | 20,017 | 89.0 | 57.0 |
|  |  | logarithmic | 0.196 | 0.286 | 0.315 |  | 64.5 | 58.7 |
|  | Gaussian | none | 0.806 | 1.461 | 0.448 | 23,549 | 68.5 | 44.4 |
|  |  | logarithmic | 0.243 | 0.285 | 0.147 |  | 55.5 | 67.3 |
|  | spherical | none | 0.623 | 1.471 | 0.576 | 22,390 | 81.4 | 43.1 |
|  |  | logarithmic | 0.218 | 0.320 | 0.319 |  | 68.5 | 64.7 |
| N | linear | none | 0.004 | 0.086 | 0.953 | 25,904 | 95.6 | 40.7 |
|  |  | logarithmic | 0.014 | 0.489 | 0.971 |  | 93.0 | 74.6 |
|  | square root | none | 0.002 | 0.077 | 0.974 | 41,211 | 95.5 | 38.6 |
|  |  | logarithmic | 0.003 | 0.445 | 0.993 |  | 89.0 | 79.3 |
|  | Gaussian | none | 0.001 | 0.083 | 0.988 | 31,791 | 68.9 | 61.5 |
|  |  | logarithmic | 0.001 | 0.444 | 0.998 |  | 99.3 | 80.1 |
|  | spherical | none | 0.001 | 0.059 | 0.983 | 28,849 | 97.9 | 47.8 |
|  |  | logarithmic | 0.006 | 0.356 | 0.981 |  | 96.2 | 78.7 |

Figure 5 shows the soil C/N rasters from the 16 variants using different interpolation parameters. An average soil C/N interpolated for 0–10 cm soil depth resulted in 11.35, while the average value for 20–30 cm soil depth was 12.60 (Table 5). The interpolation results with a logarithmic transformation had larger average value ranges, but a smaller average CV compared to no transformation. Gaussian model produced the highest value ranges and CV, representing high local variability in the interpolation results. The linear and spherical models performed similarly in terms of spatial variability, smoothing minor variabilities in soil C/N and having a CV close to 0.25. The square root model retained the lowest amount of local spatial soil C/N variability with the mean CV of 0.20. Final soil C/N results at 0–10 cm soil depth produced more variability than 20–30 cm soil depth, with a 0.05 higher average CV.

The combined scatterplots represent the correlation between interpolation results calculated using the same interpolation method. Scatterplots with Pearson's correlation coefficient are presented in the bottom-left part of the figures with no transformation, while the results of interpolation with logarithmic transformation are located in the upper-right part. Histograms of particular interpolation results are located in the middle of the figure diagonally. Combined scatterplots for 0–10 cm and 20–30 cm soil depths are shown in Figures 6 and 7, respectively. Linear and square root mathematical models produced the highest average correlation of all four interpolation sets, followed by the combination of linear and spherical mathematical models. Gaussian model produced the distinctive results, having an

average correlation coefficient to other interpolation results as low as 0.173 for 0–10 cm samples with the application of logarithmic transformation.

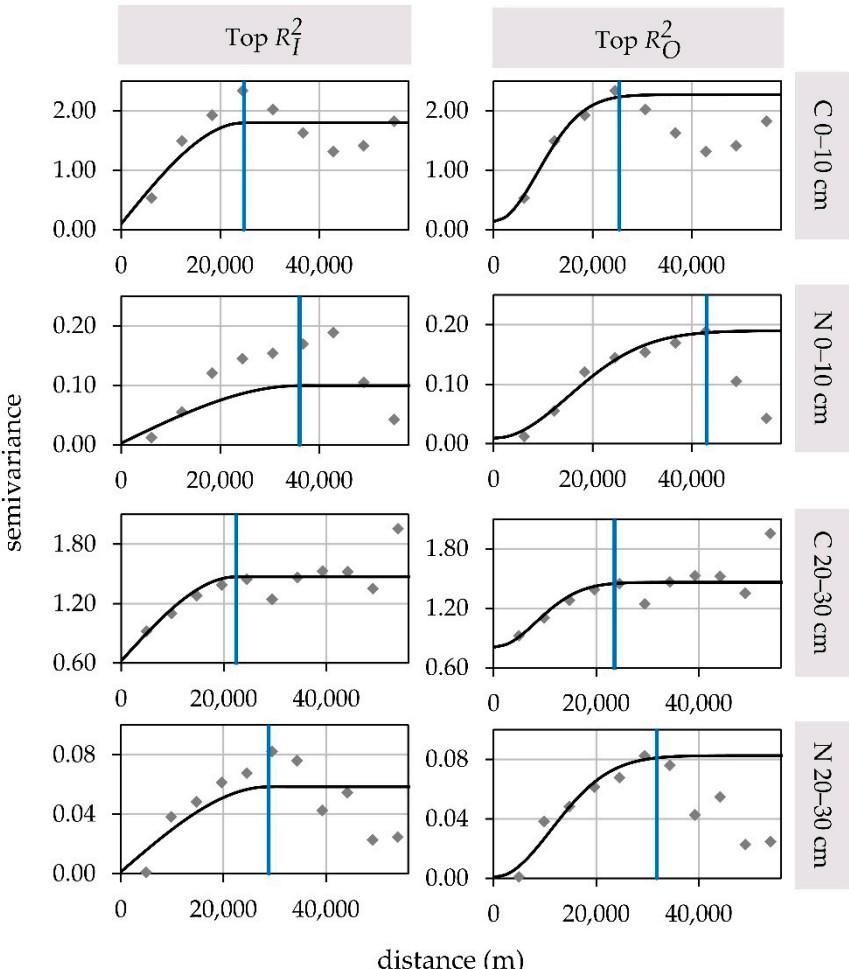

**Figure 4.** Variograms of mathematical models producing highest inner and outer accuracy.

**Table 5.** Descriptive statistics of the final soil C/N interpolation results.

| Soil Depth | Model | Transformation Method | Mean | Min | Max | CV |
|---|---|---|---|---|---|---|
| 0–10 cm | linear | none | 10.80 | 4.69 | 20.26 | 0.31 |
| | | logarithmic | 11.28 | 3.76 | 20.85 | 0.26 |
| | square root | none | 10.81 | 4.93 | 19.28 | 0.25 |
| | | logarithmic | 11.28 | 5.82 | 19.86 | 0.20 |
| | Gaussian | none | 11.45 | 3.92 | 31.70 | 0.48 |
| | | logarithmic | 12.77 | 3.88 | 37.35 | 0.46 |
| | spherical | none | 11.25 | 4.16 | 23.91 | 0.32 |
| | | logarithmic | 11.17 | 4.24 | 21.77 | 0.22 |
| 20–30 cm | linear | none | 12.05 | 4.47 | 22.20 | 0.24 |
| | | logarithmic | 11.70 | 4.44 | 26.76 | 0.25 |
| | square root | none | 12.04 | 5.53 | 22.57 | 0.14 |
| | | logarithmic | 11.54 | 4.68 | 25.74 | 0.21 |
| | Gaussian | none | 12.88 | 3.97 | 31.94 | 0.39 |
| | | logarithmic | 11.86 | 3.15 | 31.02 | 0.36 |
| | spherical | none | 12.10 | 4.17 | 26.15 | 0.23 |
| | | logarithmic | 11.62 | 3.65 | 28.60 | 0.25 |

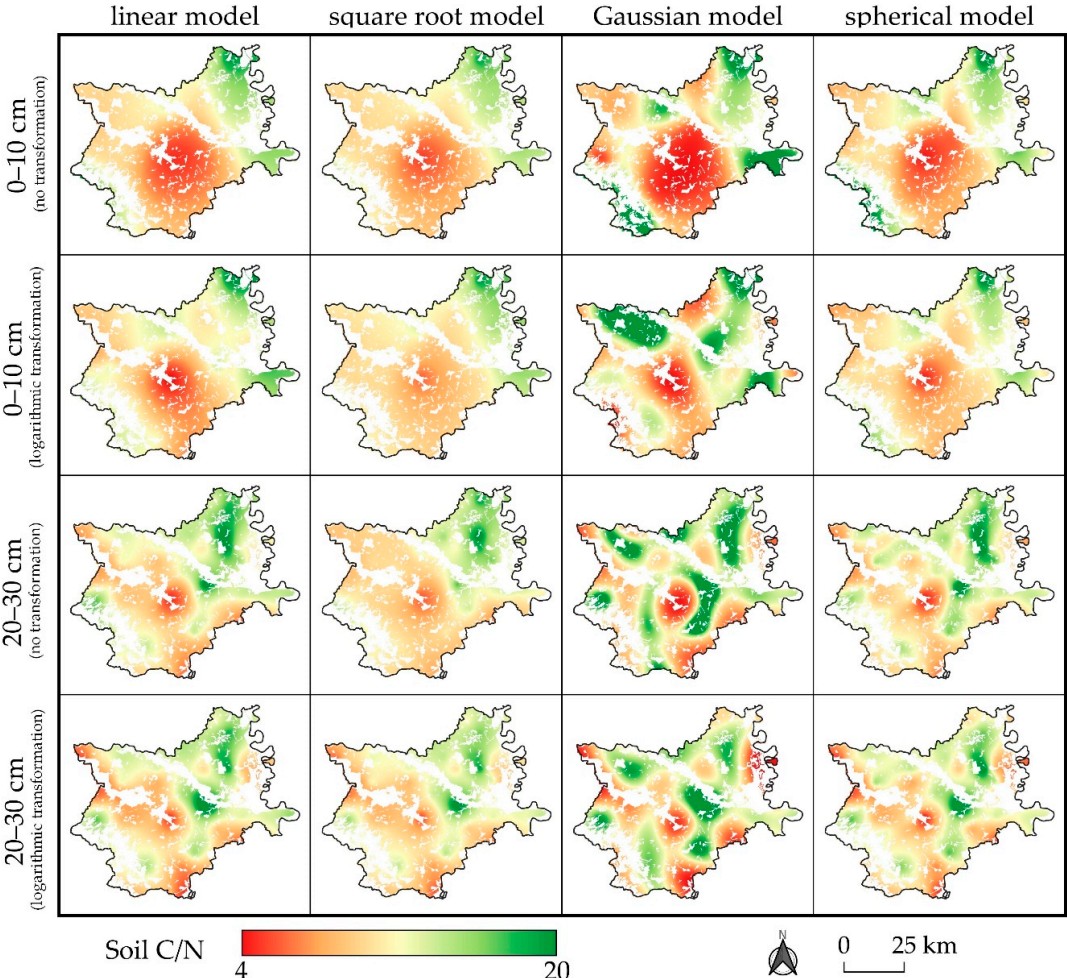

**Figure 5.** Soil carbon-to-nitrogen ratio (C/N) maps calculated using various interpolation parameters.

The soil C/N balance per major crops was evaluated according to the values displayed in Figure 8. Values from 2017 indicate a close distribution of soil C/N at 0–10 cm soil layer for maize, wheat, sunflower and rapeseed, with the mean C/N values of 12.71, 13.25, 12.42 and 12.56 to 1, respectively. Soil C/N values for soybean largely deviated from these observations, having a mean of 14.40 to 1, alongside the largest heterogeneity with the standard deviation of 7.26. These values are generally above the limit of 12:1, indicating a moderate soil N deficiency. Projections for soil C/N at 0–10 cm soil layer during 2018 and 2019 remained consistent with the 2017 values, with a slight gradual decrease of mean soil C/N for sunflower and rapeseed. Maize, wheat and rapeseed mean soil C/N values at the 20–30 cm soil layer from 2017 retained the same trend as 0–10 cm, having similar values with slightly lower standard deviation. The largest difference resulted for soybean, having a 2.62 lower mean soil C/N value at the higher soil depth. These values retained the same lower projected trend during 2018 and 2019, having the projected mean soil C/N of 11.45 to 1, which is a decrease of 0.33 compared to 2017 values. By contrast, sunflower resulted in the highest mean soil C/N value of 13.12 to 1 at the 20–30 cm soil layer during 2017 with similar projections for 2018 and 2019.

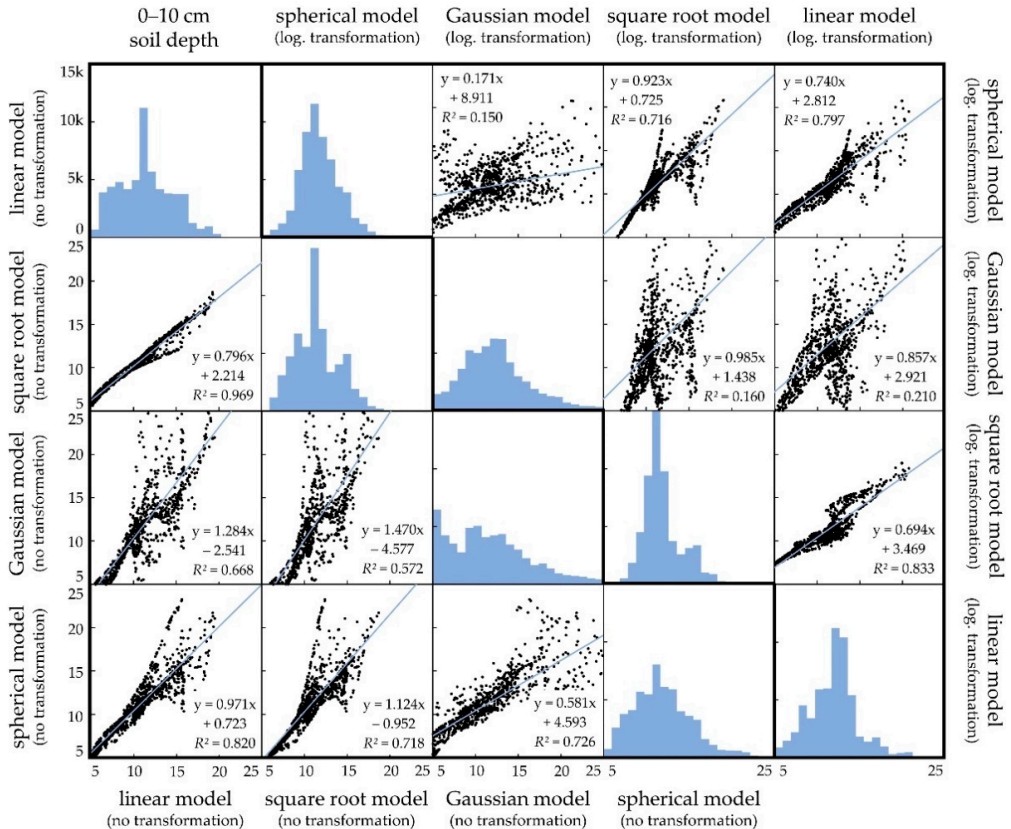

**Figure 6.** Combined scatterplot of interpolation results in 0–10 cm soil depth.

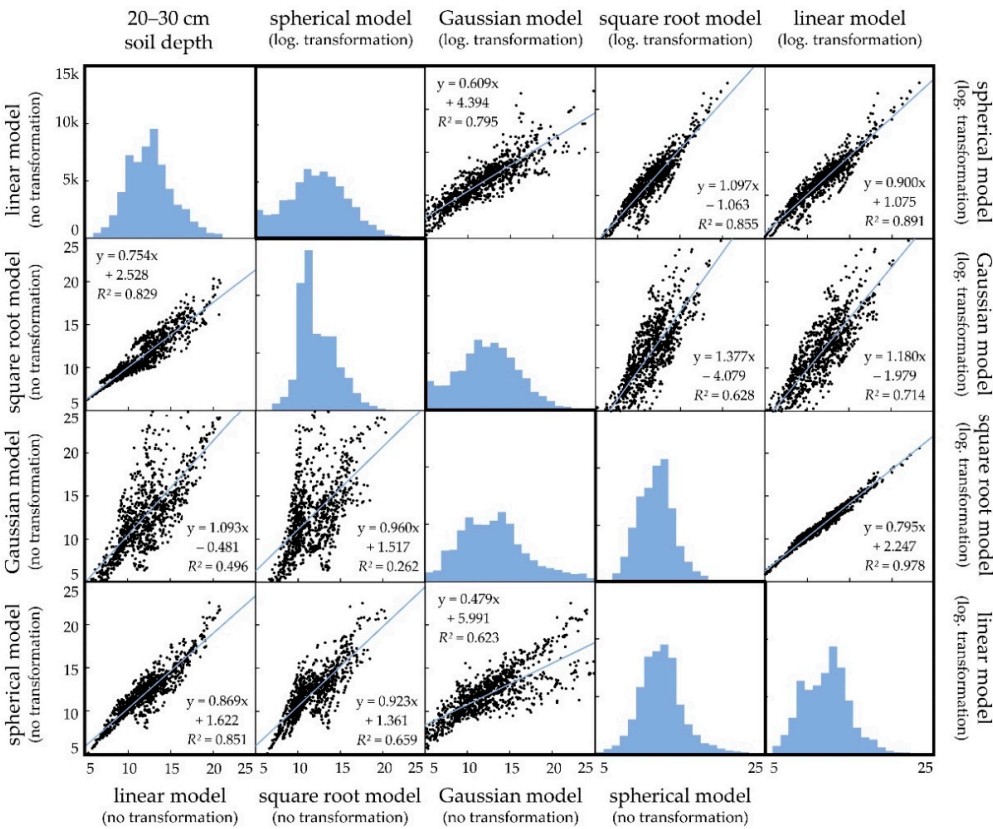

**Figure 7.** Combined scatterplot of interpolation results in 20–30 cm soil depth.

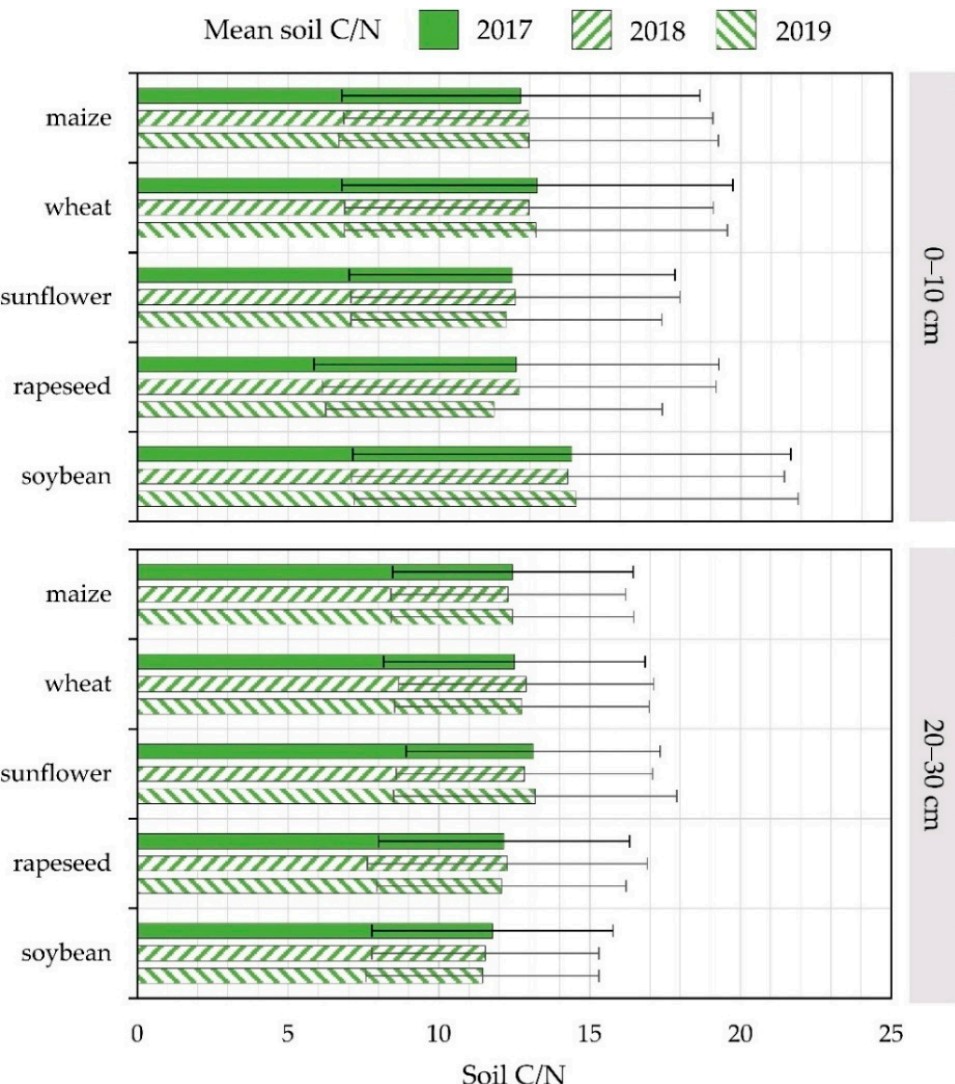

**Figure 8.** Mean and standard deviation values of soil C/N for five major crops for 2017 and projected values for 2018 and 2019.

## 4. Discussion

The interpolation accuracy and spatial variability of the predicted soil C/N were conditioned by the properties of input samples as noted in [45], confirming the hypothesis of the study. The higher spatial dependence was reflected in higher inner and outer accuracy values, based on the average spatial dependence of C and N input samples. While the application of no transformation allowed higher spatial dependence for the majority of input samples, in the case of the very low normality and stationarity, logarithmic transformation produced better results. The necessity for the objective quantification of data stationarity appeared during the evaluation of the properties of input samples, as the interpretation of Thiessen polygons heavily relies on the human subjective assessment. While logarithmic transformation had a worse effect on the preservation of ground-truth data in the interpolation results compared to no transformation, it produced considerably higher prediction accuracy, which increased with the skewness of input data. Consequentially, more different transformation types offer a potential of improving the interpolation accuracy of OK [46]. The applied procedure of the evaluation of the properties of input samples supports automatization to reduce the processing time, as every step besides stationarity evaluation was quantitatively and objectively expressed.

Kriging has been successfully applied in soil-related studies with a similar number of input samples but restricted to a county- or national-scale mapping. Schloeder et al. [47] used OK with 46 soil samples, having low stationarity and irregular sample spacing. Ding and Yu [48] successfully applied kriging on 68 samples, in combination with remote-sensing data for estimation of soil salinity. Testing of deterministic interpolation methods, such as inverse distance weighted, might be beneficial for highly skewed input samples with low sampling density [26]. The application of no transformation and a spherical mathematical model produced the highest average inner interpolation accuracy. This was reflected in the largest retention of ground-truth values from input samples in the interpolation results. Outer accuracy received much more attention in recent studies than inner accuracy [22,23,38,40], indicating an accuracy for the prediction of values at an unknown location. Logarithmic transformation and the Gaussian model were the best-performing parameters considering outer accuracy. The Gaussian model also resulted in the highest accuracy in a similar study by Fashi et al. [49], where spherical, exponential, linear and Gaussian models were tested. Besides the highest outer accuracy, the Gaussian model allowed the highest flexibility in fitting to the empirical variogram, having consistently high fitting correlation values in a wide range. These observations were made by examining a generally non-normal data with moderate stationarity, so additional evaluation using different input samples is necessary.

Average soil C/N values from the interpolation results in this study differ with the observations that soil C/N of most soils decreases with soil depth [50,51]. The lowest soil C/N values on both soil depths were observed in the central part of the study area, which is characterized by intensive agricultural production. The highest values resulted mainly in the northeastern part of the study area for both soil depths, with more heterogeneity in the case of 20–30 cm soil depth. Soil sampling density and the application of logarithmic transformation caused by varying input samples distribution had the highest effect on the spatial variability of soil C/N in the interpolation results, as noted in [27]. Evaluated mathematical models produced contrasting interpolation results, primarily in terms of the local variabilities of soil C/N. The Gaussian model, which had the highest average outer accuracy, produced the highest mean soil C/N for both soil depths. It also resulted in the highest value range and CV, which translated in the highlighting of spatial variabilities. The square root model produced the smoothest interpolation results with the smallest value range, being suitable for the mapping on a very small scale. The spherical model highlighted local spatial variabilities for most of the evaluated mathematical models, retaining the ground-truth data near to those of input samples. The selection of a mathematical model for fitting to empirical variogram had the highest impact on spatial variabilities of soil C/N, reinforcing the need for the evaluation of various alternatives for the mapping of soil features [52].

The analysis of soil C/N values at 0–10 cm and 20–30 cm soil depths determined that the current soil C/N in the study area is generally above the 12:1 value, indicating a widespread N deficiency. The major crops are commonly located at the intensively cultivated parts of the study area, which showed the greater N deficiency compared to the less cultivated area. The area of the highest soil C/N deficiency in the central part of the study area especially coincides with the area of intensive agricultural production, having an increased risk of soil degradation. Uneven distribution of soil C/N at the different soil depths could produce unexpected long-term crop development and growth, most notably for soybean and sunflower. A main observation remains for the soil C/N values for soybean at the 20–30 cm soil layer, which showed a decrease in the projections for 2018 and 2019 and the presence of vertically stratified soil C/N in the 0–10 cm and 20–30 cm soil layers. Given the non-compliance of crop rotation systems based on the Croatian Paying Agency for Agriculture, Fisheries and Rural Development data were observed at the multiple locations in the study area during 2017—2019. This is a possible cause of the lower soil C/N for soybean at 20-30 cm compared to the 0–10 cm soil layer, as well as for the large soil C/N value range for soybean, as high N soil content might occur if a legume crop was cultivated prior to soybean. These inconsistencies suggest the need for the adjustment of agricultural land management plans at the county level. This approach of regionalization of agricultural production could ensure sustainable organic farming and maximum possible yield, with possible financial gains for farmers and lower environmental contamination. One of the possible ways of accomplishing this is by the

distribution of agricultural subsidies for individual crops according to their land suitability, which in itself ensures quality climate and soil conditions for crop development [53]. GIS-based multicriteria analyses with the analytic hierarchy process (AHP) method represent a reliable and flexible solution for cropland suitability calculation globally and could assist in resolving this problem [25,54]. In addition, the constant development of remote-sensing technology offers a reliable data source for the modeling of crop suitability through vegetation indices both at macro scales using satellite imagery [55] and micro scales using unmanned aerial vehicles [56].

## 5. Conclusions

The hypothesis of the study that the selection of interpolation parameters and the properties of input samples has a major impact on the interpolation accuracy and spatial variability was confirmed. Therefore, an evaluation process of interpolation parameters and accuracy assessment is recommended in similar studies. Non-normal values distribution and moderate stationarity were observed during the evaluation of all four input samples. Logarithmic transformation negatively affected spatial dependence values compared to no transformation, which results in lower average inner accuracy. However, it enabled a more accurate prediction of values at an unknown location, represented by outer accuracy. The spherical model produced the highest ability to retain ground-truth data in the interpolation results, while the Gaussian model enabled the highest outer accuracy. The Gaussian model allowed the highest flexibility of tested mathematical models for fitting to empirical variograms, having a wide range with a high fitting coefficient of determination. Interpolated soil C/N values resulted in a higher average value at 20–30 cm soil depth, compared to 0–10 cm soil depths. The higher variability of interpolated soil C/N was observed at a 0–10 cm soil depth. This observation was contrary to some recent studies that soil C/N lowers in a deeper layer of the soil. The Gaussian model produced the highest spatial variability of predicted soil C/N, also having the largest value interval. The spherical model enabled the highest local spatial variabilities in the interpolation results, retaining the input sample values in the proximity of samples. The analysis of soil C/N distribution for five major crops in the study area (maize, wheat, sunflower, rapeseed and soybean) indicated a moderate soil N deficiency. The soil C/N values generally resulted as higher than 12:1, which differs from the optimal soil C/N range from 9:1 to 12:1 and generally indicates a moderate soil degradation risk. These observations suggest a need for the modification of present agricultural land management plans through regionalization of agricultural production to prevent further soil degradation and to ensure the best possible conditions for crop development.

**Author Contributions:** Conceptualization, M.J., D.R. and M.G.; methodology, D.R.; software, D.R.; validation, M.G.; formal analysis, M.J., D.R., S.K. and I.P.; investigation, M.J., D.R. and I.P.; resources, M.J., S.K. and I.P.; data curation, D.R.; writing—original draft preparation, D.R. and M.G.; writing—review and editing, M.J., D.R., S.K., I.P. and M.G.; visualization, D.R.; supervision, M.J. and M.G.; project administration, M.J.; funding acquisition, M.J., D.R., I.P. and M.G. All authors have read and agreed to the published version of the manuscript.

**Funding:** This research received no external funding.

**Acknowledgments:** This work was supported by the Faculty of Agrobiotechnical Sciences Osijek as a part of the scientific project "AgroGIT—technical and technological crop production systems, GIS, and land conservation". This work was supported by the University of Zagreb as a part of the scientific project: "Advanced photogrammetry and remote sensing methods for environmental change monitoring" (Grant No. RS4ENVIRO).

**Conflicts of Interest:** The authors declare no conflict of interest.

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
