# Peer review of "Geostatistical Analysis of Soil C/N Deficiency and Its Effect on Agricultural Land Management of Major Crops in Eastern Croatia"

_agronomy, doi:10.3390/agronomy10121996_

Round 1

Reviewer 1 Report

The paper presented for evaluation has a high scientific value. It is also well prepared in formal terms. In order to improve it, I present the authors with comments for consideration:

1. I propose to correct the title because plant groups (types) were not studied, but plant species. The title could be "Geostatistical analysis of soil C / N deficiency and its effect on agricultural land management of major crops types in eastern Croatia."  2. Consistently throughout the paper, the term crops should be used, not crop types.  3. Additional keywords should be used: organic matter.  4. Introduction section A few sentences can be added indicating the dependence of the quantity and quality of soil organic matter on the agronomic practices, especially fertilization and tillage, including long-term research and research on modern technologies. The results of such current research can be found, for example, in the latest editions of Agronomy 2020 (e.g. 10 (9), 10 (10)). 5. Materials and Methods section If there are available sources, the authors could provide additional information about the conditions in the research area that have a potential impact on soil organic matter, e.g. average annual rainfall, average air temperature, summer temperature, organic and mineral fertilization, dominant agricultural system, dominant soil tillage system. Please explain briefly why fewer samples were taken from the topsoil than from the 20-30 cm layer. A more detailed explanation (e.g. in the Introduction, Discussion section) requires the optimal soil C / N ratio. Shown in lines 180-182 significantly differs from the obtained results of own research. Does the quoted source refer to soil organic matter? 6. Results Some sentences can be moved to the Discussion chapter, e.g. lines 263-265. The spelling of the first lines in tables 3, 4, 5 needs to be improved (first capital letters). 7. Conclusion The conclusion should be more related to the optimal C/N ratio of the soil.

Reviewer 2 Report

The title of the manuscript (MS) deals with "Geostatistical analysis of soil C/N deficiency and its effect on agricultural land management of major crop types in eastern Croatia". The topic of this manuscript is of interest and well written and I liked reading it, great job! Just one comment. "In the "Introduction" section, this section can be improved to provide further background and include all relevant references. There is a need that you will use recent publications on the topic to make attractive your research.
